# Soil organic carbon dynamics from agricultural management practices under climate change

Tobias Herzfeld[1], Jens Heinke[1], Susanne Rolinski[1] and Christoph Müller[1]

[1]Potsdam Institute for Climate Impact Research, Member of the Leibniz Association, P.O. Box 60 12 03, 14412 Potsdam, Germany.

*Correspondence:* Tobias Herzfeld (tobias.herzfeld@pik-potsdam.de)

**Abstract.** Sequestration of soil organic carbon (SOC) on cropland has been proposed as a climate change mitigation strategy to reduce global greenhouse gas (GHG) concentrations in the atmosphere, which is in particular needed to achieve the targets proposed in the Paris Agreement to limit the increase in atmospheric temperature to well below 2 °C. We here analyze the historical evolution and future development of cropland SOC using the global process-based biophysical model LPJmL, which was recently extended by a detailed representation of tillage practices and residues management (version 5.0–tillage2). We find that model results for historical global estimates for SOC stocks are at the upper end of available literature, with ~2650 Pg C of SOC stored globally in the year 2018, of which ~170 Pg C are stored in cropland soils. In future projections, assuming no further changes in current cropland patterns and under four different management assumptions with two different climate forcings, RCP2.6, and RCP8.5, results suggest that agricultural SOC stocks decline in all scenarios, as the decomposition of SOC outweighs the increase of carbon inputs into the soil from altered management practices. Different climate-change scenarios, as well as assumptions on tillage management, play a minor role in explaining differences in SOC stocks. The choice of tillage practice explains between 0.2 % and 1.3 % of total cropland SOC stock change in the year 2100. Future dynamics in cropland SOC are most strongly controlled by residue management, whether residues are left on the field or harvested. We find that on current cropland, global cropland SOC stocks decline until the end of the century by only 1.0 % to 1.4 % if residue-retention management systems are generally applied and by 26.7 % to 27.3 % in case of residues harvest. For different climatic regions, increases in cropland SOC can only be found for tropical dry, warm temperate moist, and warm temperate dry regions in management systems that retain residues.

## 1 Introduction

To meet the targets of the Paris Agreement of 2015 to keep the increase in global mean temperature well below 2°C, and especially for the ambitious target of below 1.5°C, several negative emission technologies which remove carbon dioxide ($CO_2$) from the atmosphere have been proposed (Minx et al., 2018; Rogelj et al., 2018, 2016). At the same time as the climate is warming, the global human population is expected to increase to 9.7 billion people in 2050 and 10.9 billion by 2100 (United Nations et al., 2019), putting additional pressure on future food production systems. Food production alone has to increase by at least 50 % (FAO, 2019) or even

double by the year 2050, depending on dietary preferences, demographical trends, and climate projections, when global food demand is to be met (Bodirsky et al., 2015). Different agricultural management practices have been proposed as carbon (C) sequestration strategies to mitigate climate change and increase the quality and health of the soil by increasing soil organic carbon (SOC) content of cropland soils (Lal, 2004), which also decreases the risk of soil erosion and soil degradation (Lal, 2009).

Tillage influences many biophysical properties, such as soil temperature or soil hydraulic properties (Snyder et al., 2009), and can increase different forms of soil degradation (Lal, 1993; Kurothe et al., 2014; Cerdà et al., 2009). The potential of SOC sequestration for agricultural management practices, e.g., the effect of no-till, is debated in the scientific community (Baker et al., 2007; Powlson et al., 2014). Because tillage management is closely interrelated with residues management (Guérif et al., 2001; Snyder et al., 2009), these two practices should always be investigated simultaneously. Residue management can affect SOC stocks and soil water properties, as residues left on the soil surface can increase soil infiltration, reduce evaporation (Guérif et al., 2001; Ranaivoson et al., 2017), and add soil organic matter into the soil (Maharjan et al., 2018). Soil moisture and therefore plant productivity is also influenced by irrigation. While irrigated systems generally tend to have higher SOC stocks due to positive feedbacks on plant productivity, the feedbacks and mechanisms on SOC development are still not well understood (Humphrey et al., 2021; Emde et al., 2021). The effectiveness of irrigation systems on SOC development is influenced by climate and initial SOC stock and tends to be more effective in semiarid regions and less effective in humid regions (Trost et al., 2013).

Minasny et al. (2017) have proposed the '4 per 1000 Soils for Food Security and Climate' initiative, which targets to increase global SOC sequestration by 0.4 % per year. They argue that under best-management practices, this target rate could be even higher. This approach would translate into a 2-3 Pg C $a^{-1}$ SOC increase in the first 1 m of the soil, which is equivalent to about 20-35 % of global greenhouse gas (GHG) emissions (Minasny et al., 2017). This proposal has been criticized, as it overestimates the possible effect of SOC sequestration potential through agricultural management (de Vries, 2018; White et al., 2018). Field trials on SOC sequestration potentials show results with higher, as well as lower sequestration rates, but only represent the local soil and climatic conditions for the time of the experiment (Fuss et al., 2018; Minx et al., 2018), which reduces the likelihood for their validity on larger scales or longer time periods.

Global total SOC stocks are estimated between 1500 Pg C (excluding permafrost regions) (Hiederer and Köchy, 2011) to up to 2456 Pg C for the upper 200 cm (Batjes, 1996) and agricultural SOC stocks alone, which are subject to agricultural management, are estimated to be between 140 and 327 Pg C depending on soil depth (Jobbágy and Jackson, 2000; Zomer et al., 2017). Since the beginning of cultivation by humans approximately 12000 years ago, global SOC stocks for the top 200 cm of soil have declined by 116 Pg C because of agriculture by one estimate (Sanderman et al., 2017). Management assumptions play an important role in these estimates, e.g. Pugh et al. (2015) found that residue removal and tillage effects contribute to 6 % and 8 % of total land-use change (LUC) emissions between the year 1850 and 2012 alone, which translates into biomass and soil C losses of approx. 13.5 Pg C and 16 Pg C, respectively.

In this study, we use a modeling approach to quantify the historical development of global cropland SOC
stocks using new data for agricultural management such as manure and residues management, as well as a new
data set of the spatial distribution of tillage practices. In addition, we investigate the potential for SOC
sequestration under different climate-change scenarios on current cropland.
**2    Materials and methods**
**2.1  The LPJmL5.0-tillage2 model**
The LPJmL5.0-tillage2 model combines the dynamic phenology scheme of the natural vegetation (Forkel et al.,
2014), with version 5.0-tillage, which covers the terrestrial nitrogen cycle (von Bloh et al., 2018) and the
representation of tillage practices and residue management (Lutz et al., 2019b). The model code is available at:
https://doi.org/10.5281/zenodo.4625868 (Herzfeld et al., 2021). All organic matter pools in vegetation, litter, and
soil in LPJmL5.0-tillage2 are represented by C pools and the corresponding N pools with variable C:N ratios.
For soil carbon, the slow and fast soil pools are explicitly distributed over five soil layers (Schaphoff et al.,
2013). With the term 'SOC' we refer to the sum of all soil and litter C pools. After the harvest of crops, root
carbon is transferred to the below-ground litter pool. The incorporation of above-ground residues into the soil is
dependent on the chosen management practices. Different tillage and residue management schemes and the
accounting for direct effects of SOC on soil hydraulic properties and thus on soil organic matter (SOM)
decomposition and plant productivity have been introduced in the implementation of tillage practices in version
5.0-tillage (Lutz et al., 2019b), and are thus explicitly considered here (Fig. 1). The model accounts for an
irrigation scheme for green and blue water consumption (Rost et al., 2008) and the effects of different irrigation
systems (Jägermeyr et al., 2015). Irrigation water is dynamically calculated and coupled with the overall water
balance between soil, vegetation, and climate properties (Schaphoff et al., 2018).

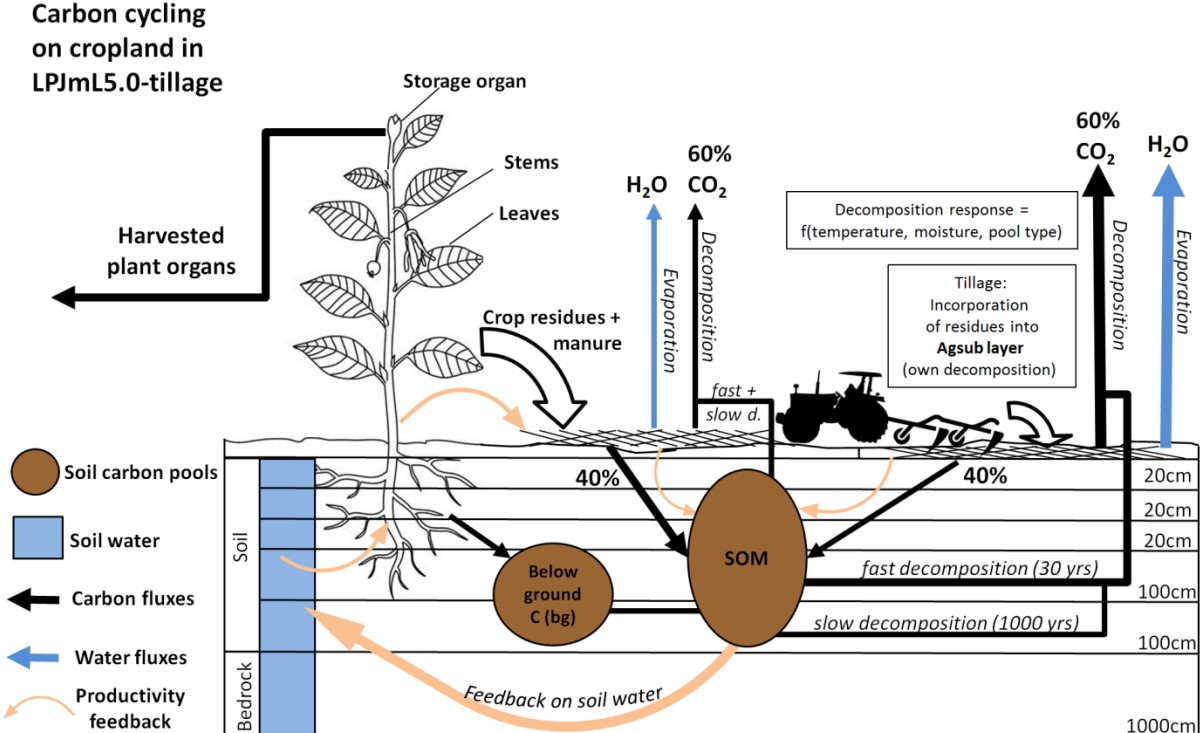


**Figure 1: Carbon cycling on cropland and productivity feedbacks from plants to residues and soil stocks and soil water, as modeled in LPJmL5.0-tillage. Arrows indicate fluxes, boxes, and circles are stocks.**

In LPJmL5.0-tillage2, the amount of carbon in biomass, which is either harvested or can be left on the field as crop residue is dependent on productivity (plant growth). Litter pool sizes are determined by the amount of biomass that is left on the field (i.e. not harvested) and the rate at which the litter is decomposed. At decomposition, the model assumes a fixed ratio of 40 % of C that is transferred from litter to the soil carbon pools; the other 60 % of C are emitted to the atmosphere as $CO_2$, as in von Bloh et al. (2018). N cycling is included in the model, explained in detail in von Bloh et al. (2018), and follows similar principles as SOC decomposition, reflecting the actual C:N ratios of the decomposing material. Applied N from manure, which is now explicitly considered in contrast to the previous model version LPJmL5.0-tillage, is assumed to consist of equal shares of mineral and organic N so that 50 % is added to the ammonium pool of the first soil layer and the rest is added to the above-ground leaf litter nitrogen pool. While manure composition is highly variable across animal type, feed, and treatment, a general ratio of 1:2 of ammonium to total N in manure is in principle supported by the ranges reported by Van Kessel and Reeves (2002). The organic leaf litter nitrogen is quickly decomposed and added to the ammonium pool of the soil. The C part of the organic manure is allocated to the leaf litter C pool (i.e. an easily degradable organic pool that can be left on the soil surface or incorporated into the soil column by tillage), with a fixed C:N ratio of 14.5 (IPCC, 2019). Total fertilizer amounts (i.e. mineral fertilizer and manure) are applied either completely at sowing or split into two applications per growing season. Manure is always applied at the first application event at sowing. Only when total combined fertilizer inputs (manure and mineral N) exceed 5 gN m$^{-2}$, half of the total fertilizer is applied in a second application as mineral

fertilizer, which is applied after 40 % of the necessary phenological heat sums to reach maturity have been
accumulated.

## 2.2 Simulation protocol

A list of the simulations carried out for this study is summarized in Table 1. An initial spinup simulation per
general circulation model (GCM) and Climate Research Unit gridded Time Series (CRU TS) climate data of
7000 years is conducted to bring SOC stocks into a dynamic pre-historic equilibrium (SP-GCM/SP-CRU), in
which the first 30 years of weather data are cyclically recycled, mimicking stable climate conditions. A second
GCM-specific spinup simulation to introduce land use dynamics starts in 1510 so that cropland older than that
has reached a new dynamic equilibrium by 1901 when the actual simulations start and land-use history is
accounted for otherwise.  Simulations were run for three groups: a) historical runs from 1901-2018 using CRU
TS Version 4.03 climate input (Harris et al., 2020) and inputs on historical management time series (which is
subject to the same spinup procedures as the GCM-specific simulations), b) historical simulations from 1901-
2005 with climate inputs from the four GCMs and historical management time series, c) future simulations using
projections of the four GCMs for the representative concentration pathways RCP2.6 (low radiative forcing) and
RCP8.5 (high radiative forcing) and four different stylized management settings: conventional tillage and
residues retained (T_R), conventional tillage and residues removed (T_NR), no-till and residues retained (NT_R)
and no-till and residues removed (NT_NR) and d) simulations as in c) but with [$CO_2$] held constant at the level
of the year 2005 (379.8 ppmv) that are used to quantify the $CO_2$ effect. All other inputs (land-use, N-fertilizer,
manure) for all future simulations were also held constant at the year 2005 values. In future simulations, we
accounted for unlimited water supply from resources available for irrigation. Additionally, the rainfed to
irrigated cropland pattern was held constant at the year 2005 pattern. An additional simulation per GCM was
conducted where all inputs, as well as management assumptions, are static after 2005. These are used to analyze
the business-as-usual case under constant land use (h_cLU). To compare the results to literature values on the
maximum potential of global SOC stocks without land use, an additional simulation with potential natural
vegetation (PNV) was conducted, where all land is assumed to be natural vegetation with internally computed
vegetation composition and dynamics.
**Table 1: Overview of the different simulations conducted for this study. For more details and purposes of the**
**simulation see text. No LU – no land use, PNV – potential natural vegetation.**

| Name | Nr. of sim. | Years | Climate input | Tillage | Residues treatment | Fertilizer | Manure | LU data-set | Description |
|---|---|---|---|---|---|---|---|---|---|
| SP_CRU SP_GCM | 5 | 7000 | CRU TS 4.03 / HadGEM2_ES, GFDL-ESM2M, IPSL-CM5A-LR, MIROC5 Repeated 1901-1930 | No LU | No LU | No LU | No LU | No LU | 7000 years PNV spin-up until 1509 to compute a pre-historic dynamic SOC equilibrium |
| SPLU_CRU SPLU_GCM | 5 | 390 | CRU TS 4.03 / HadGEM2_ES, GFDL-ESM2M, IPSL-CM5A-LR, MIROC5 Repeated 1901-1930 | First-year values of Porwollik et al. 2019 | First-year values of MADRaT | First-year values of LUH2v2 | First-year values of Zhang et al. (2017) | LUH2v2 (Hurtt et al., 2020) | 390 years spin-up until 1900 to compute the effects of LU history, which is used as the starting point for all simulations |
| h_PNV | 1 | 1901-2018 | CRU TS 4.03 1901-2018 | No LU | No LU | No LU | No LU | No LU | PNV run until 2018 (with 390 years spin-up for better comparability to LU runs), starting from SP_CRU |
| h_dLU | 2 | 1700-2018 | CRU TS 4.03 From 1700-1900 repeated 1901-1930, 1901-2018 afterward | Porwollik et al. 2019 | MADRaT (Dietrich et al., 2020) | LUH2v2 (Hurtt et al., 2020) | Zhang et al. (2017) | LUH2v2 (Hurtt et al., 2020) | Historical run with dynamic LU, starting from SPLU_CRU |
| h_cLU | 2 | 1700-2018 | CRU TS 4.03 From 1700-1900 repeated 1901-1930, 1901-2018 afterward | Porwollik et al. 2019 Static at 2005 level | MADRaT (Dietrich et al., 2020) Static at 2005 level | LUH2v2 (Hurtt et al., 2020) Static at 2005 level | Zhang et al. (2017) Static at 2005 level | LUH2v2 (Hurtt et al., 2020) Static at 2005 level | Historical run with constant land use (with 390 years spin-up as in SPLU_CRU, but with the land use pattern of 2005), starting from SP_CRU |
| h_GCM | 4 | 1901-2005 | HadGEM2_ES, GFDL-ESM2M, IPSL-CM5A-LR, MIROC5 | Porwollik et al. 2019 | MADRaT (Dietrich et al., 2020) | LUH2v2 (Hurtt et al., 2020) | Zhang et al. (2017) | LUH2v2 (Hurtt et al., 2020) | CMIP5 historical scenario runs used, starting from SPLU_GCM |
| T_R_26/85 NT_R_26/85 T_NR_26/85 NT_NR_26/85 | 64 | 2006-2099 | RCP2.6/RCP8.5 HadGEM2_ES, GFDL-ESM2M, IPSL-CM5A-LR, MIROC5 | tillage / no-till | Residues retained / residues removed | LUH2v2 (Hurtt et al., 2020) Static at 2005 level | Zhang et al. (2017) Static at 2005 level | LUH2v2 (Hurtt et al., 2020) Static at 2005 level | CMIP5 future runs with different management options, starting from h_GCM |
| TRc05_26 TRc05_85 | 16 | 2006-2099 | RCP2.6/RCP8.5 HadGEM2_ES, GFDL-ESM2M, IPSL-CM5A-LR, MIROC5 | Porwollik et al. 2019 Static at 2005 level | MADRaT (Dietrich et al., 2020) Static at 2005 level | LUH2v2 (Hurtt et al., 2020) Static at 2005 level | Zhang et al. (2017) Static at 2005 level | LUH2v2 (Hurtt et al., 2020) Static at 2005 level | CMIP5 future runs with tillage and residue management constant at 2005 level, starting from h_GCM |


## 2.3 Model inputs

We created input data sets for an explicit representation of land use, fertilizer, manure, and residue management, using the MADRaT tool (Dietrich et al., 2020). Historic land-use patterns of shares of physical cropland, also separated into an irrigated and rainfed area, as well as mineral fertilizer data (application rate per crop in $gN\ m^{-2}\ a^{-1}$) for the period of the year 1900 to 2015, are based on the Land-Use Harmonization – LUH2v2 data (Hurtt et al., 2020), which provides fractional land-use patterns for the period of 850-2015 as part of the Coupled Model Intercomparison Project – CMIP6 (Eyring et al., 2016). Manure application rates for the period 1860-2014 are based on Zhang et al. (2017) and account for organic N. With MADRaT, we were also able to produce data on crop functional type (CFT) specific fractions of residue rates left on the field (recycling shares) for the period 1850-2015. We generated data on residue-recycling shares in 5-year time steps for the period 1965-2015 and interpolate linearly between time steps to get an annual time series. Between 1850 and 1965, default recycling shares for cereals of 0.25, for fibrous of 0.3, for non-fibrous of 0.3, and no-use of 0.8 were assigned to 1850 and linearly interpolated to the values of 1965. Cereals include temperate cereals, rice, maize, and tropical cereals; fibrous crops include pulses, soybean, groundnut, rapeseed, and sugarcane; non-fibrous crops include temperate roots, tropical roots, and no-use crops include sunflower, others, pastures, bioenergy grasses, and bioenergy trees. Information on conventional tillage and conservation agriculture (no-till) management was based on Porwollik et al. (2019) for the period 1974-2010. Before 1973, conventional tillage was assumed as the default management on all cropland. We assume one tillage event after initial cultivation of natural land, independent of the tillage scenario. This assumption does not affect the results of future projections as we constrain our analysis to cropland that is already cultivated in 2005.

Historical simulations were driven using the CRU TS Version 4.03 climate input (Harris et al., 2020) from 1901 to 2018. Since this data set does not provide data before 1901, the 30-year climate from 1901 to 1930 was used repeatedly for spin-up simulations covering the period before 1901. Data on $[CO_2]$ were taken from ice-core measurements (Le Quéré et al., 2015) and the Mauna Loa station (Tans and Keeling, 2021). Future simulations from 2006-2099 used climate scenarios from four GCMs taken from Coupled Model Intercomparison Project Phase 5 (CMIP5) in bias-adjusted as provided by the ISIMIP2b project (Frieler et al., 2017; Hempel et al., 2013): HadGEM2-ES, GFDL-ESM2M, IPSL-CM5A-LR and MIROC5 for both a weak climate forcing (Representative Concentration Pathway (RCP) 2.6) and a strong climate forcing (RCP8.5) with corresponding $[CO_2]$ levels. The GCM data sets provide inputs for air temperature, precipitation, radiation, and $[CO_2]$. The historic period for these GCM-specific simulations was based on bias-adjusted data from the GCMs rather than on CRU data, to avoid inconsistencies at the transition between historic and future periods. Land-use change in the future was not analyzed in this context, as the SOC potential of the current agricultural area was the focus of this investigation so that land-use patterns after 2005 were held constant after 2005. All results are presented as averages across the ensemble of climate models per RCP, unless stated otherwise. Additional simulations with constant $[CO_2]$ for both RCP2.6 and RCP8.5 allow for the isolation of $CO_2$ fertilization effects. Conventional tillage starts in 1700. For the period 1700-1850, the residue extraction rate of the year 1850 is

assumed. The degree to which tillage affects soil properties and processes depends on the tillage intensity, which is a combination of tillage efficiency and mixing efficiency. The fraction of residues submerged (tillage efficiency) by tillage is set to 0.95. The mixing efficiency for tillage management is set to 0.9, representing a full inversion tillage practice, also known as conventional tillage (White et al., 2010). The effects of both mixing and tillage efficiency are described in detail in Lutz et al. (2019). The fraction of residues that are harvested in case of residue extraction is 70 % of all above-ground residues (with the remaining 30 % of above-ground residues and all roots left on the field). In the case without residue harvest, 100 % are left on the field and only the harvested organs (e.g. grains) are removed.

**2.4 Data analysis and metrics**

Our analysis is based on simulated changes in cropland SOC stocks as well as the contributing processes, including the turnover rate, heterotrophic respiration, litterfall, and the net primary production (NPP) of cropland areas. NPP is calculated following Schaphoff et al. (2018).

The turnover rate for cropland is calculated as:

$$mtr_{SOC,agr} = \frac{rh_{agr}}{SOC_{agr}} * 100, \tag{1}$$

with $mtr_{SOC,agr}$ as the mean turnover rate for cropland SOC (% a$^{-1}$), $SOC_{agr}$ is the SOC content for cropland (g) and $rh_{agr}$ is the heterotrophic respiration for cropland (g a$^{-1}$).

Decomposition of organic matter pools is following the first-order kinetics described in Sitch et al. (2003). Total heterotrophic respiration ($R_h$) accounts for 60 % of directly decomposed litter ($R_{h,litter}$) and respiration of the fast and slow soil pools (decomposition rate of 0.03 a$^{-1}$ and 0.001 a$^{-1}$, respectively). From the 40 % remaining litter pool, 98.5 % are transferred to the fast soil C pool and 1.5 % to the slow soil C pool:

$$R_{h,agr} = R_{h,litter,agr} + R_{h,fastSoil,agr} + R_{h,slowSoil,agrl} , \tag{2}$$

Cropland litterfall ($C_{litterfall,agr}$) in g C a$^{-1}$ is calculated by considering root, stem, and leaf carbon in dependency of residue recycling shares:

$$C_{litterfall,agr} = (C_{root,CFT} + ((C_{leaf,CFT} + C_{stem,CFT}) \cdot f_{res,CFT})) \cdot f_{cell,agr}, \tag{3}$$

with $C_{root,CFT}$ being the C pools of crop roots per CFT, $C_{leaf,PFT}$ the C pool of crop leaves per CFT, $C_{stem,PFT}$ the stems and mobile reserves per CFT, $f_{res,CFT}$ the residue fraction which is returned to the soil per CFT and $f_{cell,agr}$ the fraction of agricultural area of the cell. The h_dLU_cropland scenario uses the results from the h_dLU simulation and accounts for the cropland SOC only, by taking the cropland area at the specific point time into

account. The h_dLU_area05 scenario, on the other hand, also uses the results from the h_dLU simulation as
described in Table 1 but accounts for all the area which is either already cropland or will become cropland at any
point in time until 2005. To calculate the historical losses of SOC from land-use change in the h_dLU_area05
scenario, the fraction of SOC under PNV, which will become cropland is combined with the historical cropland
SOC parts and calculated as:
$$SOC_{LUC,t} = d_{SOC,pnv,t} \cdot (area_{agr,2005} - area_{agr,t}) + d_{SOC,agr,t} \cdot area_{agr,t}, \qquad (4)$$
where $d_{SOC,pnv,t}$ is the SOC density (g m$^{-2}$) for PNV area at time step t, which will become cropland in the
future, calculated as:
$$d_{SOC,pnv,t} = \frac{d_{SOC,cell,t} \cdot area_{cell} - d_{SOC,agr,t} \cdot area_{agr,t}}{area_{pnv,t}}, \qquad (5)$$
where $d_{SOC,pnv,t}$, $d_{SOC,cell,t}$, $d_{SOC,agr,t}$ are the SOC densities (g m$^{-2}$) for the PNV part within the cell, the density
for the entire cell, and the agricultural part within the cell, respectively, at time step t (year), $area_{pnv,t}$ and
$area_{agr,t}$ are the corresponding areas of PNV and agriculture (m$^{-2}$) at time step t and $area_{cell}$ is the area of the
entire cell, which does not change over time. We considered different climatic regions such as tropical wet,
tropical moist, topical dry, warm temperate moist, warm temperate dry, cold temperate moist, cold temperate
dry, boreal moist, and boreal dry regions, following the IPCC climate zone classification (IPCC (2006), Fig. S1
in the appendix), using averaged climate inputs for the period between the year 2000 and 2009. Polar dry, polar
moist, and tropical montane regions were excluded from this analysis, as these regions do not include any
cropland.

## 3 Model performance

Modeled global average SOC stocks (period 2000-2009 and year 2018) are compared with previous model
versions and literature estimates (Table 2). Simulated SOC stocks in LPJmL5.0-tillage2 exhibit higher SOC
content compared to the LPJmL5.0 (von Bloh et al., 2018) model version and LPJ-GUESS (Olin et al., 2015),
with total average global SOC stocks of 2640 Pg C for simulations with land use (h_dLU) and 2940 Pg C for
simulation with PNV only and no land use (h_PNV). The simulated stocks correspond well to estimates by
Carvalhais et al. (2014) for global averages but are lower for cropland SOC stocks. Total SOC stocks simulated
by LPJmL5.0-tillage2 are 2640 Pg for the entire soil column of 3 m, which are 300 Pg higher than estimates
provided by Jobbágy and Jackson (2000). Global SOC for PNV is 2580 Pg for the upper 2 m, which compares
well with estimates between 2376 Pg to 2476 Pg provided by Batjes (1996), who reported SOC stocks for the
upper 2 m of soil. Global average cropland SOC stocks between the year 2000 and 2009 as well as for the year
2018 for the entire soil column are estimated to be 170 Pg C, which is higher than estimates of 148-151 Pg C by
Olin et al. (2015). Zomer et al. (2017) reported cropland SOC stocks of 140 Pg C for the upper 0.3 m of soil,
which are higher than the cropland SOC stocks of 75 Pg C simulated for the upper 0.3 m in LPJmL. Ren et al.
(2020) reported cropland SOC stocks for the first 0.5 m of soil to be 115 Pg C for the period 2000-2010, which is
higher than cropland SOC of 95 Pg C for the upper 0.5 m in LPJmL. Scharlemann et al. (2014) conducted a
literature review on global SOC stock and found a wide range of estimates (504-3000 Pg C) and variability
across time and space and a high dependency on soil depth, with a median global SOC stock of 1460 Pg C.
Generally simulated SOC stocks by LPJmL5.0-tillage2 correspond well with literature and other model
estimates.
**Table 2: Global SOC pools (Pg C) for the LPJmL5.1-tillage2, LPJmL5.0, and LPJ-GUESS model compared to**
**literature estimates. Values are averages for the period 2000-2009, for the year 2018, and the upper 0.3, 1, and 2 m of**
**soil. PNV values are simulations with potential natural vegetation only (no land use), global SOC average includes**
**PNV and land use.**

| | Model estimates | | | Literature estimates | | | | |
|---|---|---|---|---|---|---|---|---|
| | LPJmL5.0-tillage2 (this study) | LPJmL5.0 (von Bloh et al., 2018) | LPJ-GUESS (Olin et al., 2015) | Carvalhais et al., 2014 | Batjes, 1996 | Jobbágy and Jackson, 2000 | Zomer et al., 2017 | Scharlemann et al., 2014 |
| Global SOC PNV only | 2940[1,a] 2960[2,a] 2580[b,1], 2185[c,1], 1555[d,1] | 2344[1,a] | 1671[3] | - | 2376[b,4] – 2476[b,4] | - | - | - |
| Global SOC average | 2640[1,a] 2645[2,a] 2295[b,1], 1910[c,1], 1300[d,1] | 2049[1,a] | 1668[3] | 2397[4] (1837[x] - 3257[y]) | - | 1933[b], 2344[a] | - | 1460 (504[d] – 3000[e]) |
| Cropland SOC | 170[1,a] 170[2,a] 145[b,1], 115[c,1], 75[d,1], | - | 148[3] | 327[4] (242[x] - 460[y]) | - | 210[b], 248[a] | 140[d] | - |

Values are estimates for: [a] entire soil column, [b] upper 2m of soil, [c] upper 1m of soil, [d] upper 0.3m of soil, [e] not indicated.
Year of estimate value: [1] 2000-2009, [2] 2018, [3] 1996-2005, [4] not indicated. [x] 2.5th percentile, [y] 97.5th percent
**4 Results**
**4.1 Historical development of cropland NPP and SOC stocks**
During the simulation period, cropland NPP increases in the dynamic LU simulation (h_dLU) from 0.7 Pg C a$^{-1}$
in 1700 to 4.7 Pg C a$^{-1}$ in 2018, while cropland SOC increases from 18 Pg C to a total of 171 Pg C (Fig. 2A and
2C) in the year 2018. The increase in cropland SOC can be explained by an increase in cropland area (Fig. S2B
in the appendix). During the same time, harvested C increases from 0.1 Pg C a$^{-1}$ to 2.0 Pg C a$^{-1}$. The ratio of
harvested C to cropland NPP increases with time, especially after the year 1900 (Fig. 2B), as more material is

harvested compared to cropland NPP. The aggregated SOC stock on all land that is cropland in the year 2005 declines substantially, especially after the year 1900 (red line in Fig. 2C), which reflects the decline in cropland SOC density (Fig. S2A in the appendix). We also find that cropland SOC density steadily increases between 1700 and 1950, and decreases since 1950 (Fig. S2A in the appendix). Simulations with a constant land use pattern of 2005 (h_cLU) for cropland NPP and cropland SOC show no substantial dynamics (Fig. 2A and C). These simulations are not entirely insightful, because they do not account for the historical increase in inputs, e.g. fertilizer.

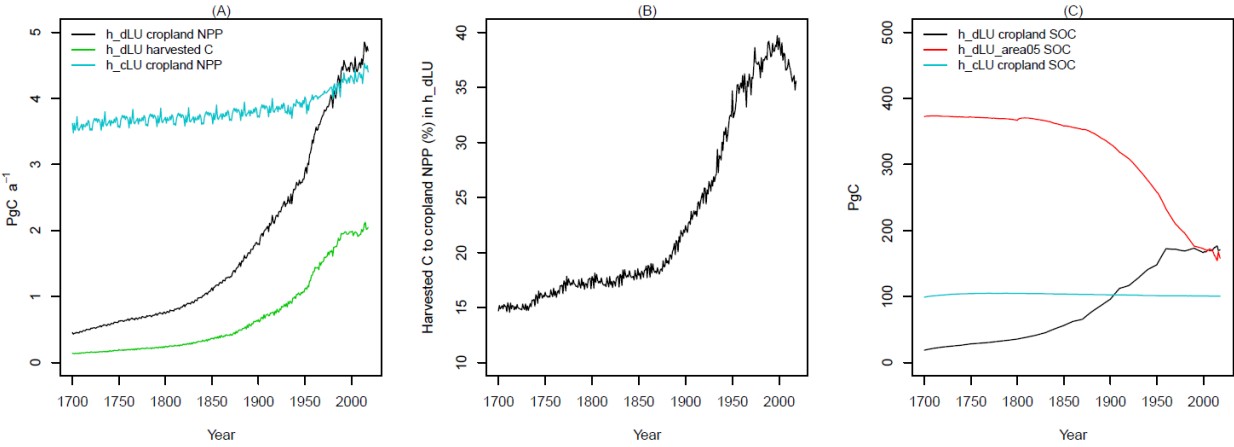

**Figure 2: Plots for cropland NPP and harvested C (A), percentage of harvested C to cropland NPP in h_dLU (B) and SOC for cropland stocks, and historical SOC losses from LUC (C) for the years 1700-2018 for simulations with transient land use (h_dLU), constant land use of 2005 (h_cLU), transient land use and SOC development from land-use change including cropland area and historical PNV area which will be converted until the year 2005 (h_dLU_area05).**

In contrast to the scenario with dynamic land use and the ones with constant land use, the h_dLU_area05 scenario describes a combination of historical cropland SOC and historical SOC of natural vegetation (calculated as described in Eq. (4) and (5)), which is or has been cropland until the year 2005. This describes the SOC dynamics of all land that is subject to the historical land-use change (LUC) (Fig. 2C). Loss of historical SOC is calculated as the difference between the years 1700 and 2018 on the land area that was cropland at any point in time (Fig. 2C, red line). Through this approach, we calculate a total historical SOC loss of 215 Pg C. Cropland SOC stocks are increasing over time (Fig. 2C, black line), reflecting the increase of cropland area. PNV has a higher SOC density, and therefore SOC stock, before the conversion to cropland (Fig. S2A in the appendix). For the calculation of SOC loss, we here only considered the area that is converted from PNV to cropland at any point in time between 1700 and 2018 in post-processing according to Eq. (4) and (5). Because SOC density is generally lower in cropland compared to PNV (Fig. S2A in the appendix), SOC is lost after conversion (Fig. 2C, red line).

**4.2 Future soil carbon development with idealized management under climate change**

Future cropland SOC stock development was analyzed considering two different radiative forcing pathways (RCPs) with four different climate scenarios (GCMs) per RCP and four idealized management assumptions (Table 2). To estimate the SOC sequestration potential on current cropland and to exclude the influence from LUC, the cropland area was kept constant at the year 2005 pattern. Results for future SOC development show that the maximum decrease in SOC stocks on current global cropland area between the year 2005 until the end of the century occurs in the scenario with no-till applied on global cropland, no residues retained, and RCP8.5 climate (NT_NR_85). Total cropland SOC loss for this scenario is evaluated as 38.4 Pg C, or 28.1 % in relative terms compared to the SOC stocks in the year 2005. All management systems, which extract residue from the field, show a strong decrease in cropland SOC stocks, independent of the climate scenario (Fig. 3B). Differences for cropland SOC development between different tillage systems as well as between the two radiative forcing pathways RCP2.6 and RCP8.5 are small. Management systems, which retain residue on the field after harvest, show the smallest reduction in cropland SOC stocks, with a maximum reduction of 5.1 Pg C (equivalent to 3.8 % decline) in the T_R_26 management system. Differences between GCM-specific climate scenarios or radiative forcing pathways (RCPs) were small in comparison to differences in residue management assumptions for SOC, turnover rates, and litterfall rates (Fig. 3) but larger than differences in assumptions on tillage systems. Only for agricultural NPP (Fig. 3A), differences in radiative forcing pathways were the main determinant of NPP dynamics, followed by GCM-specific climate scenarios.

**Table 3: Summary of absolute and relative global cropland SOC stock change between the years 2006 and 2099 for different management systems for RCP2.5 and RCP8.5 as averages across all four GCMs.**

| Management | Absolute cropland SOC change 2006 – 2099 (Pg C) | | Relative cropland SOC change 2006 – 2099 (%) | |
|---|---|---|---|---|
| | RCP2.6 | RCP8.5 | RCP2.6 | RCP8.5 |
| Tillage and residues (T_R) | -5.1 | -4.4 | -3.8 | -3.2 |
| Tillage and no residues (T_NR) | -37.6 | -38.1 | -27.5 | -27.8 |
| No-till and residues (NT_R) | -3.6 | -3.2 | -2.6 | -2.3 |
| No-till and no residues (NT_NR) | -37.8 | -38.4 | -27.7 | -28.1 |
| Tillage and residue constant as in year 2005 (TRc05) | -24.1 | -24.0 | -17.6 | -17.6 |

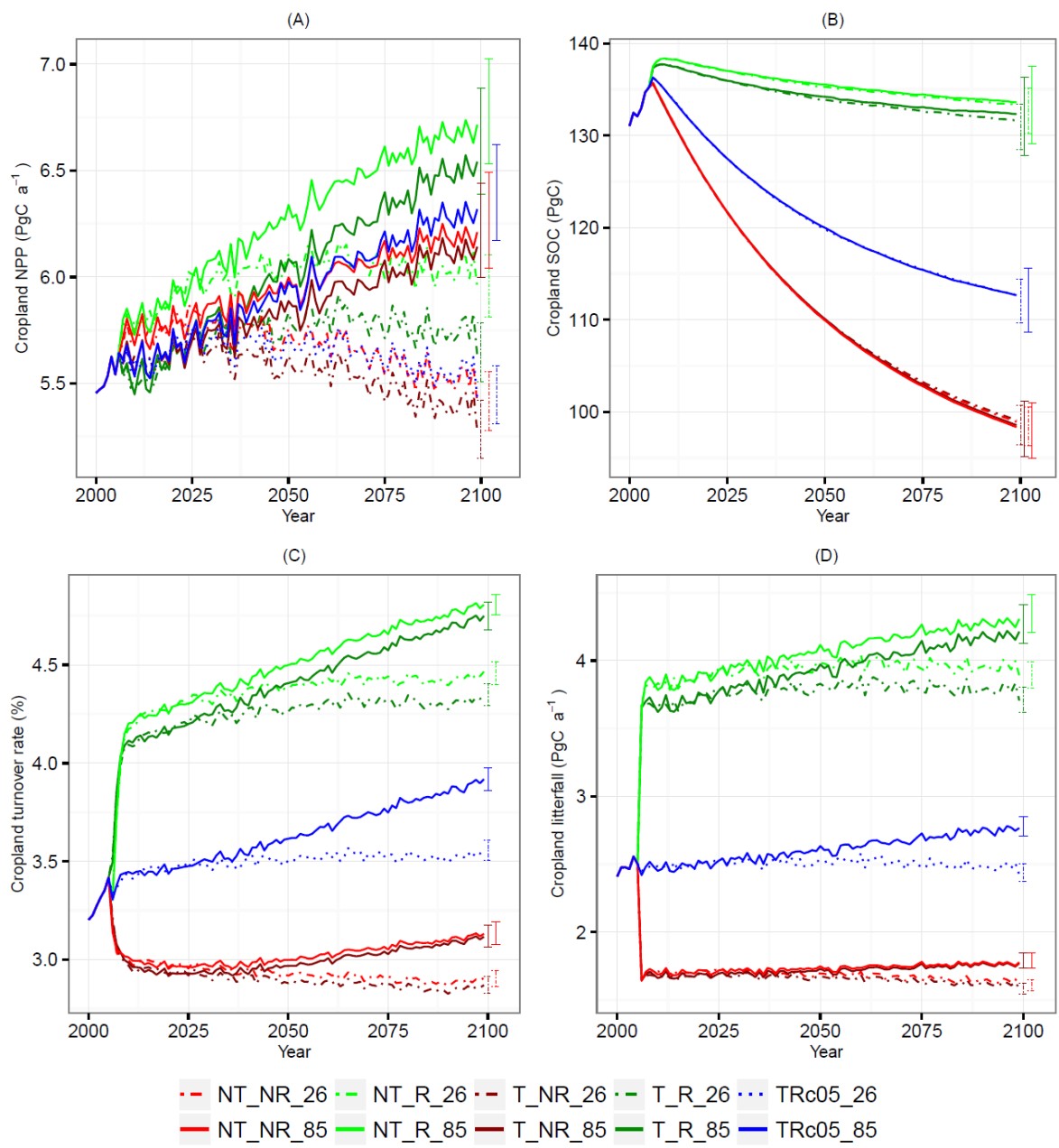

300

301

**Figure 3: Global sums for cropland for NPP (A), SOC (B), turnover rate (C), and litterfall (D) from 2000-2005 for default management inputs and from 2006-2099 under constant cropland area of 2005 for five different management scenarios and two RCPs. Presented are the mean values across all four GCMs as lines. The spread across all GCMs is depicted as bars in the year 2100. The numbers _26 and _85 describe the climate forcing RCP2.6 (e.g. TRc05_26) and RCP8.5 (e.g. TRc05_85). Green – residues retained (R), red – residues removed (NR), dashed – RCP2.6, solid – RCP8.5, light color – no-till (NT), dark color – tillage (T). Tillage and residue management held constant at 2005 level in TRc05; tillage and residues left on the field (T_R), tillage and residues removed (T_NR), no-till plus residues left on the field (NT_R) and no-till and residues removed (NT_NR). Dynamics prior to 2005 (all scenarios equal) mostly show the expansion of cropland until 2005 so that total SOC increases because the area increases. Turnover rates between 2000 and 2005 increase because decomposition rates are high on freshly deforested land.**

Stocks of cropland SOC and turnover rates (Fig. 3C) initially increase in systems that retain residues, such as
T_R and NT_R, after the change in management after the year 2005 (Fig. 3B and C), as more residual C is added
to the soil column in comparison to the historic residue removal rates (Fig. 3D). Turnover rates are higher for the
high radiative forcing pathway RCP8.5 in comparison to RCP2.6. The simulated cropland NPP (Fig. 3A) is
sensitive to the radiative forcing, as the level of NPP is higher in the high-end RCP8.5 scenario, and lower in the
lower-end RCP2.6 scenario. This is because of the strong response of NPP to $CO_2$ fertilization, which
overcompensates the climate-driven reduction in NPP (compare Fig. S3 in the appendix). NPP is less sensitive to
the assumptions on tillage practices in comparison to the effects of assumptions on residue management. The no-
till and residue system (NT_R) results in the highest NPP mainly due to water-saving effects, which are caused
by the surface litter cover, which reduces evaporation from the soil surface and at the same time increase
infiltration of water into the soil. NPP increases steadily until 2099 in RCP8.5 scenarios, because of the $CO_2$
fertilization effects (compare Fig. S3 in the appendix). In RCP2.6, NPP first slightly increases and then decreases
until the end of the century in all tillage and residue scenarios. However, the ranking of management effects is
insensitive to the radiative forcing pathway: no-till and residues (NT_R) results in the highest NPP, tillage and
no residues (T_NR) in the lowest values.

**4.3 Regional cropland SOC analysis**

Simulation results show that globally aggregated SOC stocks on current cropland decline until the end of the
century for all management systems, but there are regional differences (Fig. 4). We find that in some regions,
cropland SOC can increase until the end of the century, even though global sums indicate a total decline. For
cropland SOC density, increases between the years 2006 and 2099 can be found for T_R and NT_R management
systems for more than a third of the global cropland area, most clearly in regions in Europe, India, Pakistan,
Afghanistan, southern Chile, southern Mexico, eastern China and south-eastern USA (Fig. 4C and D).
Historically, regions which already showed an increase in cropland SOC density since 1900 until today, such as
in France or Pakistan, or a decrease, such as Canada and Argentina, tend to continue this development also in the
future (see plots in Fig. 4 for exemplary cells). In systems in which residues are not returned to the soil (T_NR
and NT_NR), global cropland SOC density change is dominated by a decline.

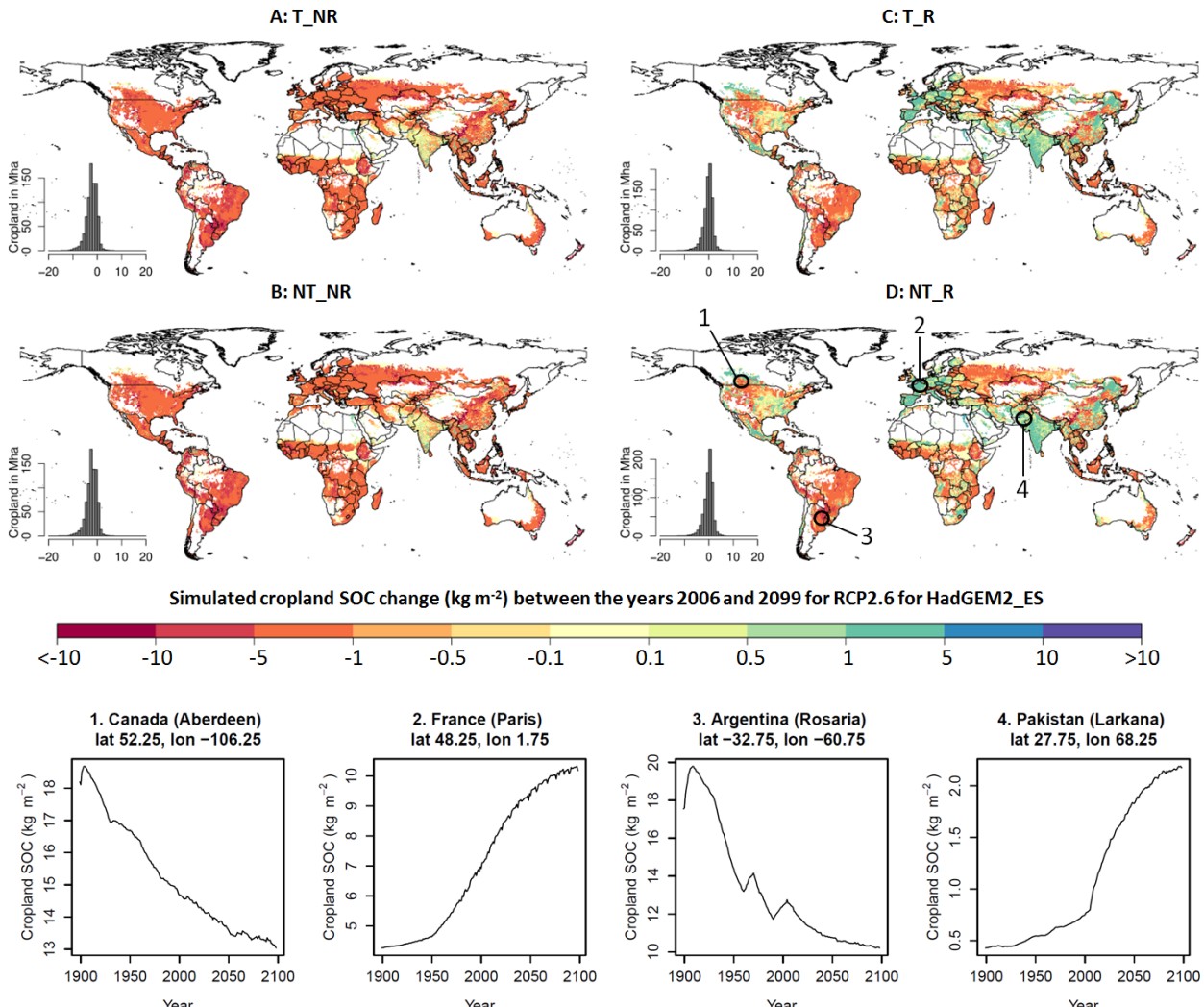

**Figure 4: Simulated cropland SOC change (kg m$^{-2}$) between the years 2006 and 2099 (kg m$^{-2}$) for RCP2.6 for GCM HadGEM2-ES for the four different management options (T_R, NT_R, T_NR, and NT_NR). The plots 1.-4. show examples of SOC development (kg m$^{-2}$) from the year 1900 to 2099 for different explanatory regions as shown on map D (NT_R). The difference maps of affected change categories between RCP2.6 and RCP8.5 are shown in Fig. 5. Maps for GFDL-ESM2M, IPSL-CM5A-LR and MIROC5, and RCP8.5 are in the appendix (Fig. S7 to S13).**

Results for different climatic regions suggest that the difference between RCP2.6 and RCP8.5 radiative forcing only plays a minor role for cropland SOC stock development (Fig. 5). Findings suggested that a positive median increase in cropland SOC density between the years 2006 and 2099 can be found in warm temperate moist, warm temperate dry, and boreal regions for RCP2.6 (GCM average) for the tillage and residue (T_R) and the no-till and residue (NT_R) management systems (Fig. 5B). The total aggregated cropland SOC change for each climate region depends on the cropland extent of the region. The smallest amounts of cropland are found in boreal moist and dry regions, which results in a total cropland SOC stock change of negligible size (Fig. 5B and D). Total increases in cropland SOC stocks can be found for both RCP2.6 (Fig. 5A and B) and RCP8.5 (GCM average) (Fig. 5C and D) for tropical dry, warm temperate moist, and warm temperate dry regions in the tillage

and residue (T_R) and the no-till and residue (NT_R) management systems. For all regions across all simulations, management systems in which residues are not returned to the soil, cropland SOC stocks decrease. The highest absolute losses of total cropland SOC stocks for these systems (T_NR and NT_NR) can be found in cold temperate dry climates, followed by tropical moist and warm temperate dry regions, which are the regions with major cropland shares.

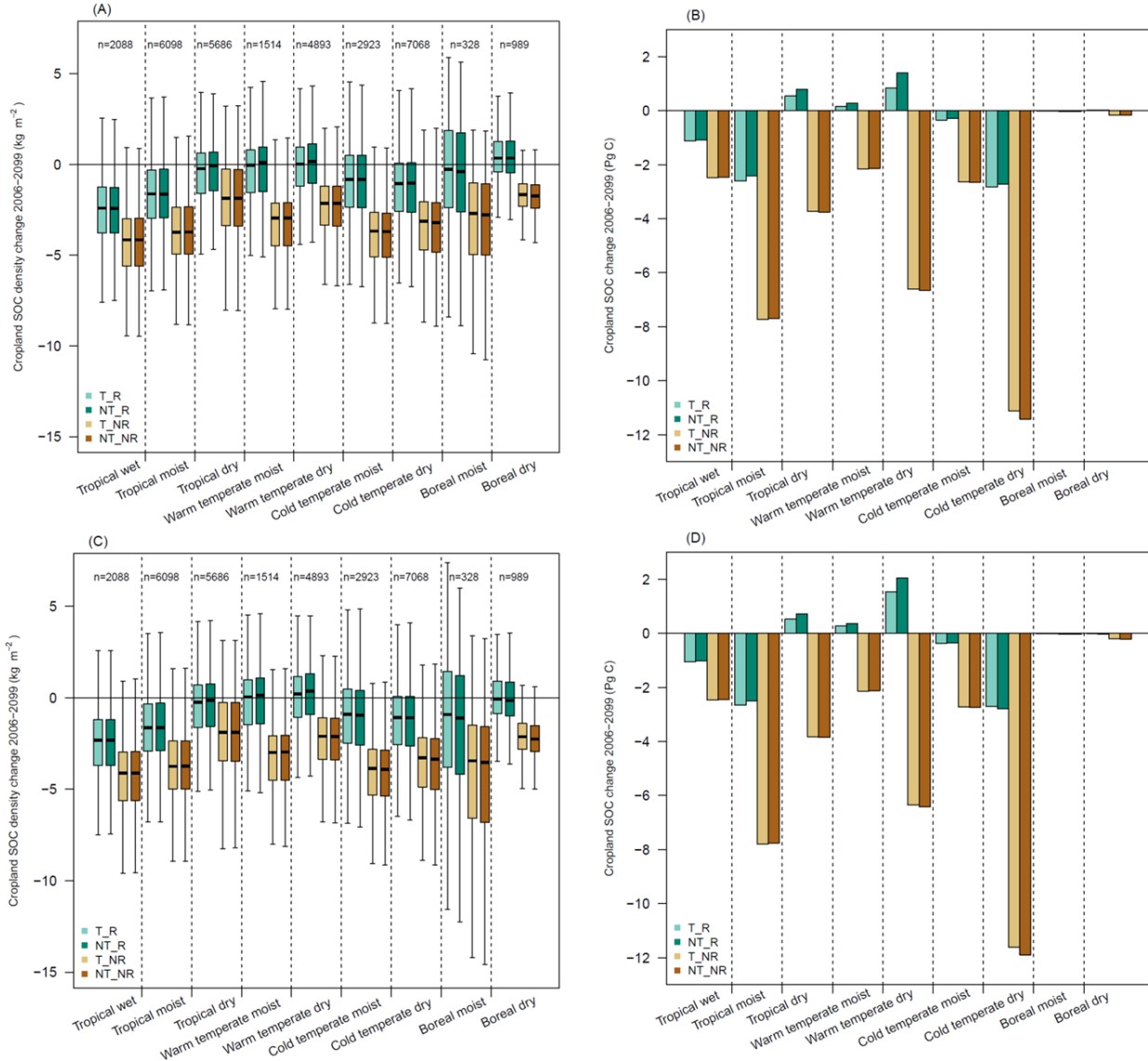

**Figure 5: Boxplots of cropland SOC density change (kg m⁻²) and bar plots of total cropland SOC change (Pg C) between the years 2006 and 2099, averaged across the four GCMs (HadGEM2_ES, GFDL-ESM2M, IPSL-CM5A-LR, MIROC5) in RCP2.6 (A and B) and RCP8.5 (C and D) for the climatic regions classified by the IPCC (2006) and the four management systems T_R, NT_R, T_NR, and NT_NR. The same plots for each GCM can be found in Fig. S5 and S6 in the appendix, n is the number of cropland cells included in each climate region.**

Regional results also indicate stronger differences between GCM-specific climate scenarios within the same
radiative forcing pathway (RCP). The highest positive cropland SOC stock response can be found for GCM
GFDL-ESM2M in both RCP2.6 and RCP8.5 for the tillage and residue (T_R) and the no-till and residue (NT_R)
systems for warm temperate dry climates, while the positive response for tropical dry and warm temperate moist
climates is lower compared to the other three GCMs (compare Fig. S5D and S6D in the appendix). Results for
the IPSL-CM5A-LR climate scenarios for both RCP2.6 and RCP8.5 generally show the most negative response
for cropland SOC density change and cropland SOC stock change, followed by HadGEM2_ES.

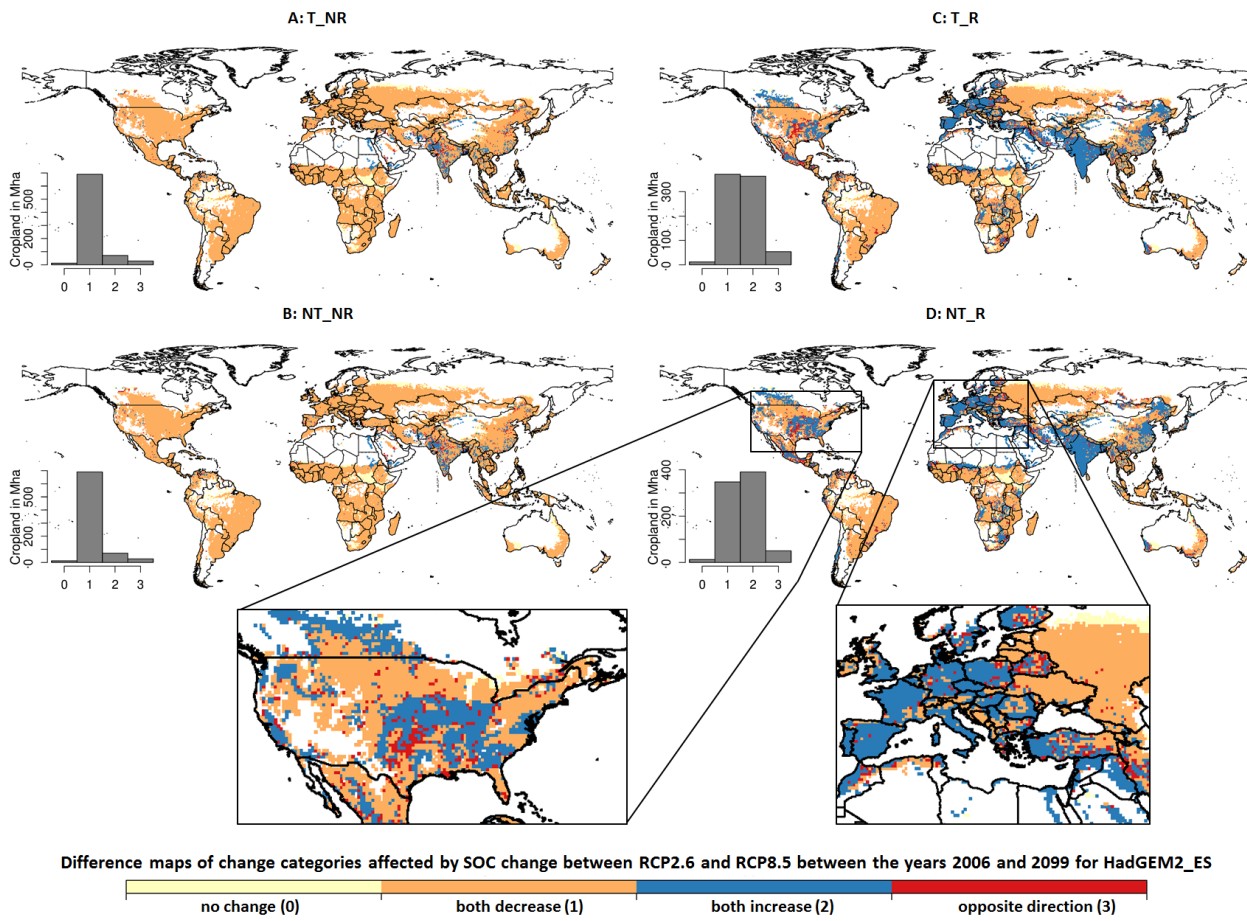

**Difference maps of change categories affected by SOC change between RCP2.6 and RCP8.5 between the years 2006 and 2099 for HadGEM2_ES**

| no change (0) | both decrease (1) | both increase (2) | opposite direction (3) |


**Figure 6: Difference maps of change categories for cropland SOC density change between both RCP2.6 and RCP8.5**
**from the year 2006 until 2099 for GCM HadGEM_ES in each management system. Orange areas indicate a reduction**
**in cropland SOC density between the years 2006 and 2099 in both RCPs, blue areas show an increase in SOC density,**
**in light yellow areas no change occurs, and for red, SOC density change occurs in opposite directions in RCP2.6 and**
**RCP8.5. The numbers in brackets (0) to (3) correspond to the categories in the histogram.**
The comparison of cropland affected in RCP2.6 and RCP8.5 indicates that most regions show effects with
the same direction of response in SOC density, so either it decreases or increases in both RCP2.6 and RCP8.5,
which is highlighted by the blue and orange regions in Fig. 6. Red cells, which indicate that the effects in both
RCPs go in the opposite direction can only be found in a few regions, e.g. the United States and Turkey. In total,
between 50 and 53 million hectares (Mha) of cropland shows the opposite directions globally for the tillage and
residue (T_R) and the no-till and residue (NT_R) systems, while this is halved (between 27 and 29 Mha) for the
tillage and no residue (T_NR) and no-till and no residue (NT_NR) management system.

## 5 Discussion

### 5.1 SOC development in the past and losses due to land-use change

Historical simulations show that the conversion of natural land to cropland has caused SOC losses of 215 Pg C
between the year 1700 and 2018 (Fig. 2C). Soil C density and NPP in natural vegetation are higher compared to
those found in croplands, which results in C losses after conversion of natural land to cropland. NPP in croplands
is often lower compared to NPP in natural vegetation, as the cultivated period is typically shorter than the
vegetative period in which natural vegetation is productive so that cultivated plants have less time to accumulate
C. Further, cropland is cultivated and crops are harvested, which results in the extraction of NPP in form of
harvested material, which leads to a further decline of SOC stocks. Cropland expansion is the main driver for
increases in total cropland SOC stocks, as cropland SOC density steadily increased since the year 1700 starting
at 7 kg m$^{-2}$ and reaching its maximum in the year 1960 at 13 kg m$^{-2}$, but since then cropland SOC density
decreased, down to 11 kg m$^{-2}$ today (Fig. S2A in the appendix). SOC density on cropland showed this trend,
even though fertilizer use increased since the 1960s, which was found to be able to promote SOC sequestration,
especially in temperate regions (Alvarez, 2005). Since the 1960s, cropland expansion has slowed down, but
global yields have, on average, more than doubled (Pingali, 2012; Ray et al., 2012; Wik et al., 2008). Ren et al.
(2020) show that historical cropland SOC increase was mainly attributed to cropland expansion, which is in
agreement with the findings here. The ratio of harvested C to cropland NPP increases with time (Fig. 2B) so that
the increase in yields does not have a positive effect on cropland SOC, as more and more C is extracted from the
soil in the form of harvested material.
It was estimated that conversion of natural land to cultivated land can result in SOC loss of up to 30 to 50
% (Lal, 2001). Sanderman et al. (2017) estimated historical global SOC losses of natural land to cropland
conversion by 133 Pg C, of which most of the losses occurred in the last 200 years.  Pugh et al. (2015) modeled
C emissions from LUC accounting for agricultural management, such as harvesting and tillage, and found
maximum C losses in vegetation and SOC by 225 Pg C since the year 1850. Le Quéré et al. (2018) also
estimated the C flux to the atmosphere due to LUC, including deforestation, to be 235 Pg C (± 95) since the year

409 1750.

### 5.2 Future cropland SOC development on current global cropland

Future SOC stocks on current cropland depend on climate and management. We find that current cropland
remains to be a source of C, even though the decline of SOC on current cropland can be reduced through
management. The most efficient measure to reduce SOC losses on cropland is residue management. In the
model, SOC is formed by C transfer from litter to the soil through decomposition fluxes (Schaphoff et al., 2018),
bioturbation, or tillage practices (Lutz et al. 2019). Residues left on the field are added to the litter C pool, where
they are subject to decomposition. Root C is added to the belowground litter pool, with a specific decomposition

according to soil temperature and moisture conditions. Stubbles and root biomass enter the litter pool after harvest, while the amount of residues extracted or retained depends on crop productivity. The addition of fresh material from crop residues increases the turnover rate in the soil, as this material is more easily decomposed than the remaining SOC stocks from the historical natural ecosystems. In the model, SOC decomposition is only driven by the temperature and moisture of the litter and soil layers, whereas the chemical composition of the residues is not taken into account. While the N content of the available material can strongly influence the decomposition and humification of residues and the formation of SOM (Hatton et al., 2015; Averill and Waring, 2018), this effect is not considered here and should be included in future model development.

The different management aspects show the same ranking in importance under both radiative forcing pathways and the changes on cropland SOC only differ slightly. Cropland SOC stocks at the end of the century vary only between those two RCPs between -0.6 % and +0.6 % for all four management systems. This is caused by a compensating effect of higher productivity by elevated $CO_2$ under RCP8.5, which counteracts the increase in turnover rates at higher temperatures (see Fig. S3 in the appendix for comparison with constant $[CO_2]$ simulations).

Even though experiments have shown that tillage can reduce SOC stocks significantly compared to no-till (Abdalla et al., 2016; Kurothe et al., 2014), tillage management only has small effects on aggregated global cropland SOC in our simulations. Tillage practices account for differences in cropland SOC stocks of 0.9 % and 1.3 % between T_R vs. NT_R in 2099 for RCP8.5 and RCP2.6, respectively, and less than 0.2 % between T_NR vs. NT_NR for both RCPs. Differences in SOC stocks on cropland between the tillage systems decrease if residues are not retained on the field. NPP responds more strongly to the tillage system, which is likely to be driven by secondary effects (e.g. no-till increases soil moisture and nutrient availability from mineralization), but shows no long-term effect on SOC stock development.

With the given complexity in responses to tillage, the application of no-tillage has been discussed ambiguously in the literature (Chi et al., 2016; Derpsch et al., 2014, 2010; Dignac et al., 2017; Powlson et al., 2014). The LPJmL5.0-tillage model is well capable of reproducing these process interactions and diversity in results (Lutz et al. 2019). Tillage systems thus need to be selected based on local conditions, but we find these to be less important than residue management. Given this dependency of the SOC accumulation potential on climatic and management conditions, there are strong regional differences in the response of SOC to changes in management. In line with Stella et al. (2019), who investigated the contribution of crop residues to cropland SOC conservation in Germany and found a decrease in SOC stocks until 2050, if residues are not returned to the soil, we find that large parts of western Europe can indeed increase the SOC stocks under management systems in which residues are retained on the field. Zomer et al. (2017) analyzed the global sequestration potential for SOC increase in cropland soils and found the highest potentials in India, Europe, and mid-west USA, results which correspond well with our findings. Also, the duration of the historical cultivation of the cropland is an important aspect in the ability to sequester C in current cropland soils. Stella et al. (2019) find the highest SOC sequestration potentials in soils with low SOC stocks (i.e. in highly degraded soils).

**5.3 Potential for SOC sequestration on cropland and recommendations for future analysis**

For the past years, there has been an ongoing debate on how much SOC can be stored in agricultural soils through adequate management as a climate change mitigation strategy (Baker et al., 2007; Batjes, 1998; Lal, 2004; Luo et al., 2010; Stockmann et al., 2013). For example, globally applied no-till management on cropland was estimated to have a SOC sequestration potential of 0.4-0.6 Gt $CO_2$ $a^{-1}$ (Powlson et al., 2014). Additionally, the sequestration of SOC can be beneficial to soil quality and productivity and minimize soil degradation (Lal, 2009, 2004). An increase in cropland irrigation can effectively influence SOC development (Trost et al., 2013; Bondeau et al., 2007). In our simulations with LPJmL5.0-tillage2, we find that on current cropland, these sequestration potentials cannot be achieved by varying tillage practices and residue removal rates, even though the residue management system is important for cropland SOC dynamics. At the same time, we account for an unlimited supply of water resources available for irrigation, reducing the constrain on SOC development by limitations from irrigation water. As such, our estimates of SOC development should tend to be optimistic in all regions where irrigation is applied, but where water resources are limiting.

There is a general uncertainty in how experimental findings can be scaled up, as e.g. demonstrated by a review conducted by Fuss et al. (2018). While process-based modeling as applied here can take environmental conditions into account and can compare different management aspects, it is still subject to various uncertainties. One crucial aspect is the history of land-use systems, including the trend in land productivity. Karstens et al. (2020, under review) show that global historical cropland SOC stocks are declining even though cropland inputs are increasing at the same time. Depending on the agricultural management option, it is argued that the maximum sequestration potential is reached after the soil has a new higher equilibrium state, which can be reached after 10-100 years, depending on climate, soil type, and SOC sequestration option (Smith, 2016). The IPCC suggests a default saturation time of the soil sink of 20 years, after which the equilibrium is reached, which then has to be maintained to avoid additional release of $CO_2$ (IPCC, 2006). Increasing cropland SOC in a first step can be achieved by adding more C to the soil than is lost by respiration, decomposition and harvest, and soil disturbance. Maintaining SOC levels on cropland after the soil has reached a new equilibrium will require the application of management strategies that do not deplete SOC. The '4 per 1000' initiative requires annual SOC sequestration on croplands of approximately 2 to 3 Pg C $a^{-1}$ in the top 1m of cropland soils, which was criticized to be unrealistic (de Vries, 2018; White et al., 2018). In this analysis, only two management options affecting SOC, tillage treatment and residues management, are considered. High SOC sequestration potentials on cropland are argued to be only achieved by applying a variety of management options, e.g. additional restoration of degraded land (Griscom et al., 2017; Lal, 2003), agroforestry (Lorenz and Lal, 2014; Torres et al., 2010), biochar (Smith, 2016), bio-waste compost (Mekki et al., 2019), which add forms of organic material which increase turnover times of SOC. A combination of these different practices is more likely to achieve higher SOC sequestration rates on cropland (Fuss et al., 2018). Management options that aim at increasing SOC may also affect yields, as they can maintain productivity and ensure yield stability (Pan et al., 2009), but reductions in SOC can also reduce yields substantially (Basso et al., 2018). Additionally, the productivity increase can come

with an even stronger increase in harvested material, as here demonstrated, which can lead to a reduction in total cropland SOC. The conversion from natural land to cropland typically causes substantial SOC losses, which stresses the need to further limit land-use expansion and thus requires an intensification of land productivity on current cropland. In our analysis, we did not account for the effects of future LUC, but projections show an increase in total cropland area in the future (Stehfest et al., 2019) so that global SOC is expected to further decline.

Further research of agricultural management practices that influence SOC development at the global scale should investigate the impact of cover crops, rotations, irrigation systems, and optimal cultivar choice per region and location (e.g. Minoli et al., 2019) and different options for cropland intensification (e.g. Gerten et al., 2020) in a more explicit manner. SOC stabilization mechanisms, such as clay mineral protection and forming of macroaggregates in no-till managed soils (Luo et al., 2016), effects of microorganisms, such as N-fixation and phosphorous acquisition from fungi and bacteria, which also regulate plant productivity and community dynamics (Heijden et al., 2008), as well as effects of soil structure (Bronick and Lal, 2005) on SOC dynamics have not been considered here or in other global process-based assessments and should be taken into account. Plants and associated root systems can reduce surface erosion and water runoff (Gyssels et al., 2005), but losses of SOC from runoff and increased erosion (Kurothe et al., 2014; Naipal et al., 2018) are not considered here either. Residues from plants can influence labile, intermediate, and stable SOC pools through the C:N ratio. Residues with high C:N ratios (e.g. straw) decomposed relatively slow and can increase SOC, but reduce N availability to the plants, while residues with low C:N decompose relatively fast and can release N to the soil through mineralization (Macdonald et al., 2018). The speed of residue decomposition can also influence the effectiveness of residues as a soil cover, with effects on soil moisture through infiltration. Impacts of biodiversity and living fauna such as microorganisms on SOC sequestration are not modeled in this analysis, even though they are recognized to have a substantial influence on the dynamics of SOC (Chevallier et al., 2001).

The implementation of such effects is desirable but needs to be assessed with respect to the process understanding, the availability of input data at the global scale, and the availability of modeling approaches (Lutz et al., 2019a). Global-scale modeling approaches, in comparison to local or regional studies, allow for the possibility to identify regional patterns related to SOC sequestration responses with the potential to foster experimental studies in areas so far not investigated, but relevant for global assessments (Luo et al., 2016; Nishina et al., 2014). They are needed to upscale findings from experimental sites so that the potential of such measures for climate change mitigation can be better understood and climate protection plans are made with better estimates.

**6 Conclusion**

In conclusion, the here analyzed agricultural management systems are not sufficient to increase global SOC stocks on current cropland until the end of the 21$^{st}$ century. The interaction of SOC sequestration and cropland productivity needs to be better disentangled. Additional C inputs from e.g. manure, cover crops, and rotations are

needed and could offset further SOC losses, but additional research on the potentials of these cropland
management options and available amounts that could be applied is needed. We find that the potential for SOC
sequestration on current global cropland is too small to fulfill expectations as a negative emission technology,
which stresses the importance to reduce GHG emissions more strictly by other means, to reach climate
protection targets as outlined in the 2015 Paris Agreement.
**Code and data availability**
The source code is available under GNU APGL version 3 license. The exact version of the code described here
and the R script used for postprocessing the data from the simulations conducted are archived under
https://doi.org/10.5281/zenodo.4625868 (Herzfeld et al., 2021).
**Author contributions**
TH and CM designed the study in discussion with JH and SR. TH conducted all the model simulations and wrote
the paper with support from CM. TH conducted the analysis and prepared all the figures with input from CM and
JH. All authors edited the paper.
**Competing interests**
The authors declare that they have no conflict of interest.
**Acknowledgments**
TH and SR gratefully thank the German Ministry for Education and Research (BMBF) for funding this work,
which is part of the MACMIT project (01LN1317A). JH thanks the BMBF for funding through the EXIMO
project (01LP1903D). TH thanks Vera Porwollik for the support in preparing input data sets and code
development. We thank the two anonymous referees for their helpful comments to improve the paper.

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
