# Peer review of "Soil organic carbon dynamics from agricultural management"

_Earth System Dynamics, 2021_

## Author Comment (AC1)

Dear Editor and Referee,

We would like to thank the anonymous referee for the valuable and constructive comments that will help to improve the manuscript. In the following, we will address all the comments by referee #1 and respond to these. Line numbers in our response refer to the marked-up version of the manuscript, which we will upload as soon as the option is available.

**Referee #1:**

**Preface:** The manuscript describes a modelling study to analyse historical and future fluxes of soil organic carbon and its sequestration potential on global croplands. The authors describe in detail the various simulation steps to calculate historical SOC stocks and projected SOC changes under different management and climate scenarios. The estimated SOC stocks are compared to previous estimates. Model results are sensitive to assumptions about different amounts of residue remaining in the field, while different tillage and climate change scenarios have a small impact on estimated SOC stocks. The authors reach the conclusion that carbon sequestration potential as a climate change mitigation strategy is low and that SOC stocks are mainly declining by the end of the century. This article is of high scientific value and I recommend it for publication without any substantial revision. In general, this manuscript is well written and simple enough to understand. I have noticed some model input values whose origin I do not understand. It would improve the understanding of the model if the following were more clearly explained.

*Response to preface:* Thank you for the positive response to the manuscript and for suggesting it for publication after some edits and clarification.

**Referee comment 1:** Line 82-83 – What is the 40/60 ratio based on that determines how much C is released into the soil or emitted into the atmosphere?

*Response to comment 1:* This is a standard model assumption in LPJmL, originally set to 30/70 by Sitch et al. (2003) and changed to 40/60 when the nitrogen cycle was introduced (von Bloh et al. 2018). We have added the reference accordingly to line 99.

**Referee comment 2:** Line 87 – Similar to above, where does the 50% come from?

*Response to comment 2:* As the sentence states, this is an assumption made. Manure composition is highly variable across animal type, feed, storage, and treatment. Van Kessel and Reeves (2002) report 0.9 to 9.5 kg total N $m^{-3}$ and 0.3 to 4.7 kg $NH_4$-N $m^{-3}$ across different dairy manure types, which in principle supports a 50/50 split. In the model, the organic manure N in the litter pool quickly decomposes to $NH_4$ and this parameter is thus not too important for plant N availability or N losses. We have added a short sentence and the reference in the main text in lines 104-107.

**Referee comment 3:** Line 156 – What does tillage intensity set to 0.9 mean?

*Response to comment 3:* Thank you for pointing to this. This was falsely described as tillage intensity. It was meant to describe the mixing efficiency of tillage, which was set to 0.9. The tillage intensity in the model is a combination of tillage efficiency and mixing efficiency, as described in Lutz et al. (2019). A mixing efficiency for tillage management of 0.9 represents a full inversion tillage practice, also known as conventional tillage, as described in White et al. (2010).
We have now edited this section in the manuscript and briefly describe tillage efficiency and mixing efficiency and refer to Lutz et al. (2019) in lines 178-183.

**Referee comment 4:** Line 300 – "a third" instead of 1/3

**Response to comment 4:** We have changed this in the manuscript.

**Referee comment 5:** Finally, spelling out the management scenarios in the text would improve readability rather than using the abbreviations (T_NR, NT_NR, etc.), although I understand that this can be handled according to personal preference. In addition, spelling out the management scenarios in Table 3 would save time for the reader who may not have the abbreviations in mind.

**Response to comment 5:** For better readability, we have spelled out most of the abbreviations in the updated manuscript, as well as in Table 3.

**References:**

Lutz, F., Herzfeld, T., Heinke, J., Rolinski, S., Schaphoff, S., Bloh, W. von, Stoorvogel, J. J., and Müller, C.: Simulating the effect of tillage practices with the global ecosystem model LPJmL (version 5.0-tillage), 12, 2419–2440, https://doi.org/10.5194/gmd-12-2419-2019, 2019.

Van Kessel, J. and Reeves, J.: Nitrogen mineralization potential of dairy manures and its relationship to composition, Biol Fertil Soils, 36, 118–123, https://doi.org/10.1007/s00374-002-0516-y, 2002.

White, J. W., Jones, J. W., Porter, C., McMaster, G. S., and Sommer, R.: Issues of spatial and temporal scale in modeling the effects of field operations on soil properties, 10, 279–299, https://doi.org/10.1007/s12351-009-0067-1, 2010.

---

## Author Response (AR1)

Point-by-point response on the reviews for the manuscript: "SOC sequestration potentials for agricultural management practices under climate change" by Herzfeld et al. (2021) - ESD 2021-35

01st September 2021

Dear Editor and Referees,

We would like to thank the two anonymous referees for the valuable and constructive comments that helped to improve the manuscript. We have addressed all the comments and updated the manuscript accordingly. In the following, we will respond to all the comments by the two referees and explain the changes made to the manuscript (response in green). The line numbers in our response refer to the marked-up version of the manuscript.

**Referee #1:**

**Preface:** The manuscript describes a modelling study to analyse historical and future fluxes of soil organic carbon and its sequestration potential on global croplands. The authors describe in detail the various simulation steps to calculate historical SOC stocks and projected SOC changes under different management and climate scenarios. The estimated SOC stocks are compared to previous estimates. Model results are sensitive to assumptions about different amounts of residue remaining in the field, while different tillage and climate change scenarios have a small impact on estimated SOC stocks. The authors reach the conclusion that carbon sequestration potential as a climate change mitigation strategy is low and that SOC stocks are mainly declining by the end of the century. This article is of high scientific value and I recommend it for publication without any substantial revision. In general, this manuscript is well written and simple enough to understand. I have noticed some model input values whose origin I do not understand. It would improve the understanding of the model if the following were more clearly explained.

*Response to preface:* Thank you for the positive response to the manuscript and for suggesting it for publication after some edits and clarification.

**Referee comment 1:** Line 82-83 – What is the 40/60 ratio based on that determines how much C is released into the soil or emitted into the atmosphere?

*Response to comment 1:* This is a standard model assumption in LPJmL, originally set to 30/70 by Sitch et al. (2003) and changed to 40/60 when the nitrogen cycle was introduced (von Bloh et al. 2018).
**Changes in the manuscript:** We have added the reference accordingly to line 100.

**Referee comment 2:** Line 87 – Similar to above, where does the 50% come from?

*Response to comment 2:* As the sentence states, this is an assumption made. Manure composition is highly variable across animal type, feed, storage, and treatment. Van Kessel and Reeves (2002) report 0.9 to 9.5 kg total N m$^{-3}$ and 0.3 to 4.7 kg $NH_4$-N m$^{-3}$ across different dairy manure types, which in principle supports a 50/50 split. In the model, the organic manure N in the litter pool quickly decomposes to $NH_4$ and this parameter is thus not too important for plant N availability or N losses.
**Changes in the manuscript:** We have added a sentence and the reference in the main text in lines 105-108.

**Referee comment 3:** Line 156 – What does tillage intensity set to 0.9 mean?

*Response to comment 3:* Thank you for pointing to this. This was falsely described as tillage intensity. It was meant to describe the mixing efficiency of tillage, which was set to 0.9. The tillage intensity in the model is a combination of tillage efficiency and mixing efficiency, as described in Lutz et al. (2019). A mixing efficiency for tillage

management of 0.9 represents a full inversion tillage practice, also known as conventional tillage, as described in White et al. (2010).
**Changes in the manuscript:** We have edited this section in the manuscript and briefly describe tillage efficiency and mixing efficiency and refer to White et al. (2010) and Lutz et al. (2019) in lines 180-185.

**Referee comment 4:** Line 300 – "a third" instead of 1/3

*Response to comment 4:* Thank you for highlighting this.
**Changes in the manuscript:** We have changed this in the manuscript in line 339.

**Referee comment 5:** Finally, spelling out the management scenarios in the text would improve readability rather than using the abbreviations (T_NR, NT_NR, etc.), although I understand that this can be handled according to personal preference. In addition, spelling out the management scenarios in Table 3 would save time for the reader who may not have the abbreviations in mind.

*Response to comment 5:* Thank you for the suggestion.
**Changes in the manuscript:** For better readability, we have spelled out most of the abbreviations in the updated manuscript, as well as in Table 3.

**Referee #2:**

**Preface:** This manuscript used a processed-based model to simulate the cropland SOC stocks change historically and in the future under different climate scenarios. The historical simulations of global and cropland SOC stocks are comparable with previous studies. The future projections with various agricultural practices show that residue management has a greater impact on SOC stocks as compared to tillage. This study provides important insight on preferred management to maximize cropland SOC storage under climate change. My main concern is that the future simulations did not include the impact of irrigation. Besides increasing temperature and CO2, climate change also has a strong impact on regional precipitation, in turn, the soil moisture and vapor pressure deficit. Studies have shown that soil moisture has a strong impact on the global carbon cycle (e.g. Humphrey, V., et al. (2021). "Soil moisture-atmosphere feedback dominates land carbon uptake variability." Nature 592(7852): 65-69.). In this study, the authors only mentioned croplands were separated into the irrigated and rain-fed areas in the historical simulations (line 122-124). Did the future simulations use the same irrigation management as the year 2015? Can the authors at least clarify their method of determination of the irrigated and rain-fed areas and add discussion on the impact of irrigation practice and its interaction with other managements on cropland SOC stocks. I look forward to reading a revised version of this manuscript.

*Response to preface:* Thank you for your review and the useful recommendations to improve the manuscript. To reply to your main concern, irrigation and water interactions with the biosphere are indeed important drives of productivity and impacts plant growth, and irrigation is considered in our future simulations. LPJmL accounts for an irrigation scheme of blue and green water consumption (Rost et al., 2008), as well as different irrigation systems, such as sprinkler, surface, and drip irrigation (Jägermeyr et al., 2015). Irrigation water partitioning is dynamically calculated in coupling to the modeled water balance and climate, soil, and vegetation properties (Schaphoff et al., 2018). In our future simulations, we do not account for adjustments in irrigation systems or irrigated areas that could be implemented in response to changing climate conditions, but as we generally do not account for future land-use change, we keep the irrigation system, as well as the rainfed/irrigated share, constant throughout the future simulation period as in the year 2005. The irrigated and rainfed area was separated based on the Land-Use Harmonization – LUH2v2 data, as described in lines 144-147. On irrigated cropland, we simulate potential irrigation, which assumes unlimited water resources available for irrigation rather than constraining irrigation to surface discharge, which underestimates irrigation water withdrawals that also tap groundwater resources.
Still, we also believe that the role of irrigation systems on cropland SOC dynamics should be considered in future analysis. As previous studies using LPJmL have shown, for example in semiarid regions, irrigation is responsible for an increase in soil carbon (Bondeau et al., 2007).

**Changes in the manuscript:** We updated the model description and explained the irrigation scheme of LPJmL in more detail in lines 89-92 and describe the irrigation setting in the simulation protocol in lines 132-134. We added a discussion on the irrigation effectiveness to the lines 469-470, lines 472-475, and added 'irrigation systems' as a field for further research in line 511.

**Referee comment 1:** The "SOC sequestration potential" in the title seems to be misleading. Sequestration indicates SOC accumulation even under climate change as long as we use proper management. However, the results of this study show that the global SOC stocks decrease under all climate scenarios and management. I think using something like "SOC stock dynamics" is more appropriate.

*Response to comment 1:* Thank you for the suggestion. Because we not only analyze the changes in SOC stocks but also changes in SOC density, we decided to additionally remove "stock" from the title and spell out SOC for clarity.
**Changes in the manuscript:** We changed the title to "Soil organic carbon dynamics from agricultural management practices under climate change".

**Referee comment 2:** The impact of various agriculture practices, such as residue management, tilling, and irrigation should be described in the introduction to set up for the results and discussion.

*Response to comment 2:* Thank you.
**Changes in the manuscript:** We have extended the introduction by a description of tillage, residue management, and irrigation effects in lines 39-51.

**Referee comment 3:** Line 115 – the potential natural vegetation data need a reference.

*Response to comment 3:* We do not use any external data on potential natural vegetation (PNV), but LPJmL can compute the natural vegetation composition and dynamics internally. The PNV simulations simply ignore the land-use input data and assume that there is no land use.
**Changes in the manuscript:** We have added additional text to clarify this to line 138-139.

**Referee comment 4:** Line 237-238 – this description of h_dLU_area05 is quite confusing. Can the authors describe this scenario in the method?

*Response to comment 4:* We are sorry for the confusion caused.
**Changes in the manuscript:** Please see the response to comment 5 below.

**Referee comment 5:** Line 240-243 – the h_dLU_area05 scenario described here is more clear, but it still should be described in the method and be listed as one of the scenarios in Table 1, because it is related to the main conclusion that cropland SOC stock decrease over history.

*Response to comment 5:* Thank you, we are aware that the calculation of the results of the h_dLU_area05 might confuse. While 'h_dLU' refers to the simulation set up as described in Table 1, the extension '_cropland_SOC' and '_area_05' refers to the post-processing of model results. In the '_area_05' case, the results are calculated as described in Eq. (4) and (5). This was not clear in the original text and has been updated now.
**Changes in the manuscript:** We have improved the description of the h_dLU_area05 scenario analysis in the method section in lines 208-214. We also added the calculation reference to Eq. (4) and (5) in the results section in lines 274-275 for more clarity on how the results are calculated.

**Referee comment 6:** Line 248-249 – this sentence needs some editing. Did the authors mean the calculation of the actual decrease in SOC stocks from LUC considered areas that were converted to cropland at any time over the entire period (1700-2018)?

***Response to comment 6:*** This is indeed the case. For the calculation of SOC loss from LUC, we only considered the areas which are actually converted from PNV to cropland at any point in time between 1700 and 2018. Because SOC density is generally lower on cropland compared PNV, SOC is reduced.
**Changes in the manuscript:** We edited the sentence in lines 281-285 for clarification.

**Referee comment 7:** Line 272 – what lead to the sudden jump of SOC, turnover rate, and litterfall between 2000 and 2005 in all management scenarios?

***Response to comment 7:*** Cropland SOC increases in all management scenarios between 2000 and 2005 as land-use patterns are dynamic and only kept constant after 2005. So this reflects the increase in cropland area between 2000 and 2005. The same holds true for turnover rates, where prior to 2005 newly deforested land is added to the cropland pool, where decomposition rates are high because of the unusually high amount of fresh material after deforestation. There is no jump in litterfall prior to 2005, and after 2005 the different management practices show strong effects on the SOC pool, turnover rate, and litterfall.
**Changes in the manuscript:** We have added text to the caption of Figure 3 for clarification in lines 316-318.

**Referee comment 8:** Line 379-382 – the mechanism of SOC forming should be better described and referenced here. The number of residues that can be retained on cropland also depends on both the quantity and quality of residues. The priming effect is not always positive. Since the authors discussed the compensating effect of higher productivity and turnover rates in the following paragraph, the effect of temperature on organic matter decomposition should be described here to set up the following discussion.

***Response to comment 8:*** Thank you.
**Changes in the manuscript:** We have updated this section in lines 422-428 and added a more detailed description of how C is transferred to the soil and SOC formation occurs. We now also discuss that the model only considers temperature and moisture as drivers of the decomposition but not the quality of residues left on the field in lines 430-434.

**Additional changes in the manuscript:** In lines 423-427 and lines 444-446 of the original manuscript, we refer to Karstens et al. (2020, under review), who initially reported an increase in historical SOC stocks. After a revision of their methodology, they now have reported an update on their findings, which suggests the opposite. We have considered these changes in their results and updated this section in the new version of the manuscript accordingly in lines 479-481 and removed the statement in lines 502-503. We have also adjusted the land-use data reference in the last three rows of Table 1, which falsely referenced 'Hurtt et al. (2017)' instead of 'LUH2v2 (Hurtt et al. 2020)'.

**References:**

Bondeau, A., Smith, P. C., Zaehle, S., Schaphoff, S., Lucht, W., Cramer, W., Gerten, D., Lotze-Campen, H., Müller, C., Reichstein, M., and Smith, B.: Modelling the role of agriculture for the 20th century global terrestrial carbon balance, 13, 679–706, https://doi.org/10.1111/j.1365-2486.2006.01305.x, 2007.

Jägermeyr, J., Gerten, D., Heinke, J., Schaphoff, S., Kummu, M., and Lucht, W.: Water savings potentials of irrigation systems: global simulation of processes and linkages, 19, 3073–3091, https://doi.org/10.5194/hess-19-3073-2015, 2015.

Karstens, K., Bodirsky, B. L., Dietrich, J. P., Dondini, M., Heinke, J., Kuhnert, M., Müller, C., Rolinski, S., Smith, P., Weindl, I., Lotze-Campen, H., and Popp, A.: Management induced changes of soil organic carbon on global croplands [preprint], in review, 1–30, https://doi.org/10.5194/bg-2020-468, 2020.

Lutz, F., Herzfeld, T., Heinke, J., Rolinski, S., Schaphoff, S., Bloh, W. von, Stoorvogel, J. J., and Müller, C.: Simulating the effect of tillage practices with the global ecosystem model LPJmL (version 5.0-tillage), 12, 2419–2440, https://doi.org/10.5194/gmd-12-2419-2019, 2019.

Rost, S., Gerten, D., Bondeau, A., Lucht, W., Rohwer, J., and Schaphoff, S.: Agricultural green and blue water consumption and its influence on the global water system, 44, W09405, https://doi.org/10.1029/2007WR006331, 2008.

Schaphoff, S., Bloh, W. von, Rammig, A., Thonicke, K., Biemans, H., Forkel, M., Gerten, D., Heinke, J., Jägermeyr, J., Knauer, J., Langerwisch, F., Lucht, W., Müller, C., Rolinski, S., and Waha, K.: LPJmL4 – a dynamic global vegetation model with managed land – Part 1: Model description, 11, 1343–1375, https://doi.org/10.5194/gmd-11-1343-2018, 2018.

Van Kessel, J. and Reeves, J.: Nitrogen mineralization potential of dairy manures and its relationship to composition, Biol Fertil Soils, 36, 118–123, https://doi.org/10.1007/s00374-002-0516-y, 2002.

White, J. W., Jones, J. W., Porter, C., McMaster, G. S., and Sommer, R.: Issues of spatial and temporal scale in modeling the effects of field operations on soil properties, 10, 279–299, https://doi.org/10.1007/s12351-009-0067-1, 2010.